# Improving CLIP Adaptation by Breaking Tail Alignment for Source-Free Cross-Domain Few-Shot Learning

**Shuai Yi** [1 2]  **Yixiong Zou**[✉ 2]  **Yuhua Li**[✉ 2]  **Ruixuan Li**[✉ 2]

## Abstract

Vision-Language Models (VLMs) such as CLIP demonstrate strong zero-shot generalization, but their performance significantly degrades in cross-domain scenarios with scarce target-domain training data (Cross-Domain Few-Shot Learning, CDFSL). In this paper, we focus on the target-domain few-shot finetuning in the CLIP-based CDFSL task. Prevailing finetuning paradigms uniformly align all image patch tokens with their corresponding textual embeddings. However, we find a counterintuitive phenomenon: actively pushing away certain low-similarity image tokens, termed "tail tokens", from their textual embeddings consistently improves target-domain performance. We delve into this phenomenon and provide a novel interpretation: under great domain shifts and scarce training data, the model can hardly extract semantic information from visual inputs; therefore, the common belief of alignment is valid only for tokens already containing sufficient semantic information; for tail tokens, forcing the alignment would lead to excessive overfitting to the scarce training, while breaking the alignment is more useful. Motivated by this, we propose Adaptive Tail-Head Alignment (ATHA), a novel fine-tuning strategy for CLIP that transforms the conventional uniform alignment paradigm to an adaptive alignment paradigm, with both alignment strengthening and weakening. Extensive experiments on four challenging CDFSL benchmarks validate our state-of-the-art performance. Our code is available at https://github.com/shuaiyi308/ATHA.

[1]Institute of Artificial Intelligence, Huazhong University of Science and Technology, Wuhan, China [2]School of Computer Science and Technology, Huazhong University of Science and Technology, Wuhan, China. Correspondence to: Yixiong Zou <yixiongz@hust.edu.cn>, Yuhua Li <idcliyuhua@hust.edu.cn>, Ruixuan Li <rxli@hust.edu.cn>.

*Proceedings of the $43^{rd}$ International Conference on Machine Learning*, Seoul, South Korea. PMLR 306, 2026. Copyright 2026 by the author(s).

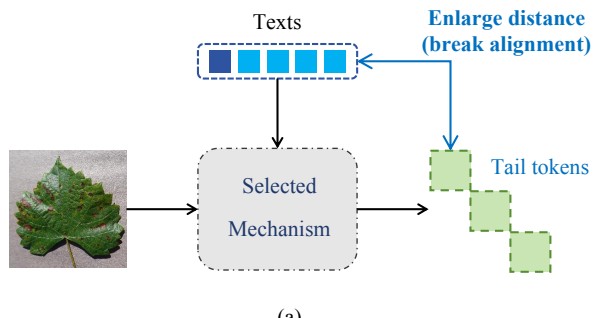

(a)

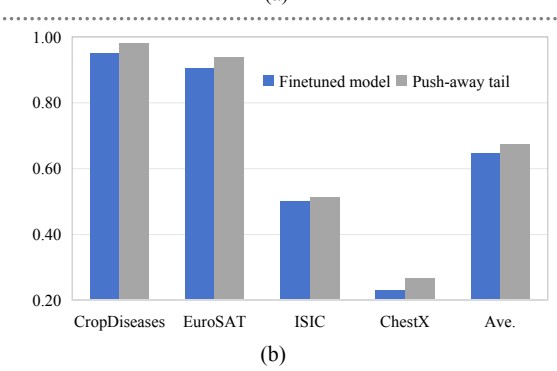

(b)

*Figure 1.* **The Counterintuitive Efficacy of Pushing Away Tail Tokens.** (a) We identify tokens with the lowest semantic similarity to any class text (Tail Tokens) and increase their distance during target-domain few-shot finetuning. (b) We find that this operation consistently improves target-domain performance, contradicting the prevailing paradigm of vision-text alignment and validating our insight: in cross-domain adaptation, selective repulsion of harmful alignment is as crucial as strengthening useful ones.

## 1. Introduction

Vision-Language Models (VLMs) such as CLIP achieve remarkable zero-shot generalization by learning semantically aligned image-text representations through contrastive pre-training (Radford et al., 2021; Zhao et al., 2023). To adapt these models to downstream tasks, a prevalent paradigm is to strengthen the alignment between visual tokens (e.g., image patches) and their corresponding textual concepts during finetuning, where generally better alignment improves task-specific representation (Xu et al., 2024a). This paradigm is also widely applied in Cross-Domain Few-Shot Learning (CDFSL) (Xu et al., 2024b), where the goal is to adapt the

source-domain model to target domains with large domain shifts through only a few target-domain training samples.

In this paper, we focus on the target-domain finetuning of the CLIP-based CDFSL task (Yi et al., 2025b), and find a counterintuitive phenomenon that challenges this prevailing alignment view: during target-domain few-shot fine-tuning, actively pushing away low-similarity image tokens from texts can consistently improve target-domain performance. As illustrated in Fig. 1a, our operation begins by identifying "Tail Tokens", patches with the lowest semantic relevance to any target class, and is followed by subtracting text embeddings from less similar tail tokens, which serves as an effective way to reduce vision-text similarities. This operation leads to consistent performance gains across diverse target domains, surpassing standard fine-tuning (Fig. 1b). In other words, we find that explicitly breaking the alignment between tail tokens and text embeddings is beneficial, contrary to the common belief of alignment-oriented finetuning.

We then delve into this phenomenon for an interpretation. Through extensive analysis, we find that during the target-domain finetuning, forcing tail tokens to align with textual embeddings amplifies the overfitting to target-domain training data, represented as absorbing excessive training-data-specific information and therefore reducing the similarity between source and target domains. Conversely, by strengthening the alignment of head tokens (patches with the highest semantic relevance to target classes), the overfitting is effectively reduced, leading to higher performance. This demonstrates that under great domain shifts and scarce training data, the model can hardly extract semantic information from visual inputs; therefore, if we simply force tokens with scarce semantic information (tail tokens) for alignment, the model cannot learn to align them in a rational way, but can only memorize them, leading to overfitting. In this case, breaking the alignment can conversely help the learning. In other words, for the target-domain finetuning in CDFSL, the common belief of alignment is valid only for tokens already containing sufficient semantic information (head tokens); for tail tokens, breaking the alignment is more useful.

Based on this insight, we propose Adaptive Tail-Head Alignment (ATHA), a novel fine-tuning strategy for CLIP that transforms the conventional uniform alignment paradigm to an adaptive alignment paradigm. At its core, ATHA implements an adaptive alignment policy for tokens and layers: during forward propagation in every ViT block, it dynamically identifies and asymmetrically modulates visual tokens based on their semantic relevance to target classes. Specifically, head tokens are pulled closer to their corresponding text embeddings via layer-wise learnable addition of text embeddings, while tail tokens are pushed away from text embeddings by learnable subtraction of text embeddings. Extensive experiments on four challenging CDFSL bench-

marks validate the efficacy of our approach.

Our contributions are summarized as follows:

- We identify and analyze a counterintuitive phenomenon in target-domain finetuning of CDFSL: actively pushing away certain image tokens from text embeddings improves target-domain performance, challenging the prevailing uniform alignment paradigm.

- We provide a novel interpretation, showing that for the target-domain finetuning in CDFSL, the common belief of alignment is valid only for tokens already containing sufficient semantic information (head tokens); for tail tokens, breaking the alignment is more useful.

- We propose a simple yet effective token- and layer-adaptive alignment method that dynamically strengthens or suppresses token alignment via learnable scaling parameters to control adding or subtracting text embedding for effective manipulation of alignment.

- Extensive experiments validate our state-of-the-art performance across multiple benchmarks.

## 2. Related Work

**Cross-Domain Few-Shot Learning** (CDFSL) addresses the challenging scenario where models must rapidly adapt to novel target domains with only a handful of labeled examples (Tseng et al., 2020; Li et al., 2022). Existing approaches generally follow two paradigms: meta-learning methods (Fu et al., 2022; Hu & Ma, 2022) that train models through episodic tasks to acquire fast adaptation capabilities, and transfer learning methods (Zhou et al., 2023; Yi et al., 2025b) that enhance generalization during source pre-training. A particularly practical but difficult extension, Source-Free CDFSL (SF-CDFSL) (Xu et al., 2024b), prohibits access to source domain data during adaptation, placing exclusive emphasis on target-domain fine-tuning strategies. Recent studies have explored various fine-tuning approaches for vision-language models like CLIP in this setting (Zanella & Ben Ayed, 2024; Xu et al., 2024a; Yi et al., 2026), but typically follow the intuition that all image patches should be aligned with textual descriptions to bridge domain gaps. In contrast, we discover a counterintuitive phenomenon where actively pushing-away certain patches significantly improves performance, challenging this prevailing paradigm.

**Cross-modal alignment**, which seeks to establish dense correspondences between local visual tokens and linguistic concepts, has become a pivotal direction for enhancing vision-language models. Representative works pursue this through various mechanisms: SPARse Fine-grained Contrastive Alignment (SPARC)(Bica et al., 2024) computes language-grouped vision embeddings as the weighted

average of patches and obtains local alignment through a fine-grained sequence-wise loss; Patch Aligned Contrastive Learning (PACL)(Mukhoti et al., 2023) advances open-vocabulary segmentation by aligning image patches with category embeddings via contrastive learning; and Contrastive Localized Pre-Training strengthens local correspondence through region-aware contrastive objectives (Chen et al., 2024). A fundamental premise across these methods is that comprehensive alignment of all relevant tokens universally improves representation. However, in CDFSL, this premise is critically challenged due to huge domain shift and scarce datasets. We find a novel phenomenon ignored by previous works: breaking the alignment of certain tokens leads to consistent performance gains in CDFSL, and propose a novel adaptive alignment method to deal with CDFSL problems.

## 3. Why Breaking Tail Alignment Helps

### 3.1. Preliminaries

**Source-Free Cross-Domain Few-Shot Learning:** In this work, we focus on the target-domain finetuning of the cross-domain few-shot learning (CDFSL) task (source-free CDFSL)(Yazdanpanah & Moradi, 2022; Xu et al., 2024a). This task is defined over a target domain dataset $\mathcal{D}_T$. For finetuning and evaluation, we adopt the episodic paradigm: an episode $\mathcal{E}$ is constructed by first sampling $N$ classes from $\mathcal{D}_T$, then drawing $K$ labeled images per class to form the support set $\mathcal{S}$, and sampling a disjoint set of $M$ images from the same $N$ classes to form the query set $\mathcal{Q}$. This forms an $N$-way $K$-shot classification task(Oh et al., 2022; Li et al., 2022). The goal is to rapidly adapt a model using only the few labeled examples in $\mathcal{S}$ and then predict the labels for samples in $\mathcal{Q}$. Crucially, in the source-free setting(Xu et al., 2024b), no data from the pretraining (source) domain is accessible during this adaptation phase; the model must rely solely on its pretrained parameters (e.g., a CLIP model) and the few-shot support set $\mathcal{S}$ from the novel target domain.

**Token Alignment in CLIP Finetuning:** For classification, a prompt template (e.g., "a photo of a class") is used to generate textual descriptions for each class(Zhao et al., 2023). Let $\mathbf{r}_k$ be the tokenized prompt for class $k$, the text encoder produces its embedding $\mathbf{t}_k = F_t(\mathbf{r}_k)$. Similarly, each image $\mathbf{x}_i$ is processed by the visual encoder $F_v$ to obtain the normalized visual embedding $\mathbf{f}_i = F_v(\mathbf{x}_i)$. The cross-entropy loss for the task defined over the image-text similarity scores is:

$$\mathcal{L}_{\text{cross}} = -\frac{1}{N} \sum_i \log \frac{\exp(\text{sim}(\mathbf{f}_i, \mathbf{t}_i)/\tau)}{\sum_j \exp(\text{sim}(\mathbf{f}_i, \mathbf{t}_j)/\tau)}, \quad (1)$$

where $\text{sim}(\mathbf{f}_i, \mathbf{t}_j) = \frac{\mathbf{f}_i^\mathsf{T} \mathbf{t}_j}{|\mathbf{f}_i||\mathbf{t}_j|}$ denotes cosine similarity and $\tau$ is the temperature coefficient.

**Pushing Away Patch Tokens in Vision Transformer:** In

CLIP's Vision Transformer, an input image $\mathbf{x} \in \mathbb{R}^{H \times W \times 3}$ is divided into $L$ non-overlapping patches, each projected to a $D$-dimensional token. After adding the [CLS] token and positional embeddings, we obtain the initial visual token sequence $\mathbf{V}^{(0)} \in \mathbb{R}^{(L+1) \times D}$. Through $N_l$ transformer layers, these tokens are transformed as $\mathbf{V}^{(l)} = \text{TransformerBlock}^{(l)}(\mathbf{V}^{(l-1)})$, where $l \in [1, N_l]$. The final [CLS] token $\mathbf{v}_{\text{CLS}}^{(N_l)}$ is used for classification. During fine-tuning, a common practice is to align each patch token with textual embeddings, but we reveal this uniform alignment strategy is suboptimal under large domain gaps.

Specifically, our pushing-away operation is conducted by

$$v_t' = v_t - \beta \cdot t, \quad (2)$$

where $\beta$ is a hyperparameter, $t$ is the closest class names to the tail token $v_t$. Since $v_t' \cdot t = (v_t - \beta \cdot t) \cdot t = v_t \cdot t - \beta \cdot t^2 < v_t \cdot t$, the vision-text similarity effectively decreases, breaking the tail token alignment[1].

### 3.2. How Pushing Away Tail Tokens Affects the Model

In Fig. 1, our empirical finding demonstrates that actively repelling low-similarity tail tokens from text embeddings yields consistent gains in target-domain finetuning. To understand this phenomenon, we study the distribution of *token-wise similarities* by plotting the cosine similarity between each visual token (image patch) and its most relevant class text embedding on target-domain images.

To this end, we systematically compare the vision-text similarity distributions across four critical models on all four benchmark datasets[2]: (1) the pretrained model, (2) the model fine-tuned under the standard paradigm, and (3) the model fine-tuned with only the tail token repulsion operation ("Push-away tail"). The comparative analysis, visualized in Fig. 2, reveals a clear narrative of alignment dynamics under domain shift.

The pretrained model (Fig. 2a) exhibits an ideal multimodal alignment state in the source domain. Its similarity distribution shows a clear hierarchical structure: a few key tokens (head tokens) demonstrate high similarity to relevant class text embeddings, forming a distinct right peak, while the vast majority of tokens (tail tokens) maintain a low baseline similarity, forming a prominent left valley.

When we transfer the pretrained model directly to the target domains (Fig. 2bcd), the distribution changes significantly, which becomes flatter and more concentrated, losing the sharp hierarchical structure observed in the source domain. This indicates that, due to the large domain shift, image tokens are generally less sensitive to the textual descriptions of target domain classes and lack discriminativity.

---

[1]We also compare with loss-based methods in Experiments.
[2]Refer to Appendix for results on more datasets.

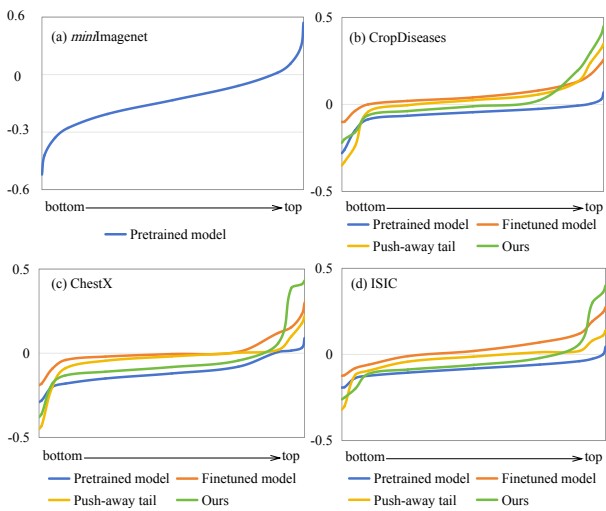

*Figure 2.* Vision-text similarity distribution of the pretrained CLIP model evaluated on the source domain, exhibiting an ideal, hierarchical structure where a few discriminative head tokens show high similarity to class texts, while the majority of tail tokens remain at a low similarity baseline. On the target domains, the pretrained model (transferred directly) shows a flat, less discriminative curve due to domain shift. Standard fine-tuning causes a global upward shift, pulling both head and tail tokens closer to text. The Push-away-tail method suppresses the alignment of tail tokens while still enhancing head tokens, indicating that inhibiting the alignment of tail tokens contributes majorly to the performance gain.

The model after standard fine-tuning (Fig. 2bcd) shows a markedly different distribution. The entire curve significantly moves upwards, indicating a global increase in vision-text similarity, not only pulling head tokens closer but also pulling tail tokens toward the text.

In the "Push-away tail" model (Fig. 2bcd), the distribution presents a hybrid characteristic. The left portion of the curve (representing tail tokens) is successfully suppressed and shifted back towards lower similarity compared to the standard fine-tuned model. Notably, the right part (head tokens) is not diminished; in fact, it shows a further upward shift compared to the pretrained model. This indicates that even by solely repelling tail tokens, the model can align head features while effectively restraining the tail tokens. In other words, inhibiting the alignment of tail tokens is a primary driver for the performance gain.

### 3.3. Aligning Tail Tokens Leads to Overfitting

Then, we study why aligning tail tokens harms the performance. As tail tokens represent tokens with the least similarity to the class texts, it is harder to pull these tokens close to texts compared with head tokens. Also, since domain shifts as well as scarce training data also prevent the model from learning effective patterns in target domains, we hypothesize that the model actually cannot learn to align

tail tokens with texts at all, i.e., it just memorizes the tokens in the target-domain training data for forcely aligning tail tokens, leading to overfitting.

To verify this hypothesis, we follow (Kornblith et al., 2019) to use Centered Kernel Alignment (CKA) similarity to measure the domain similarity between the source and target domains, where higher similarity indicates lower domain discrepancy. If such overfitting really happens, features extracted by the model would contain excessive information specific to target-domain training data, leading to an abnormally low domain similarity[3]. Also, breaking tail alignment would increase such an abnormally low domain similarity.

As shown in Fig. 3, we compare CKA similarities under three model configurations: (1) pre-trained model, (2) standard fine-tuned model, and (3) the model fine-tuned with only the tail token repulsion operation ("Push-away tail"). After standard fine-tuning, the CKA similarity decreases significantly, suggesting that tokens absorb substantial target-domain information. With the "Push-away tail" method applied to break the tail alignment, the CKA similarity increases, verifying our hypothesis.

Building on our earlier analysis, we further note that the head tokens are aligned in both the source domain and the "push-away tail" method, and this alignment correlates with improved performance on target domains. That is, although the domain shifts and scarce training data prevent the model from aligning vision and texts, since head tokens already contain sufficient semantic information (as they are closer to semantic texts), the model can still learn useful information during the finetuning, making the alignment of head tokens beneficial to the performance. This drives us to our conclusion: **the common belief of alignment is valid only for tokens already containing sufficient semantic**

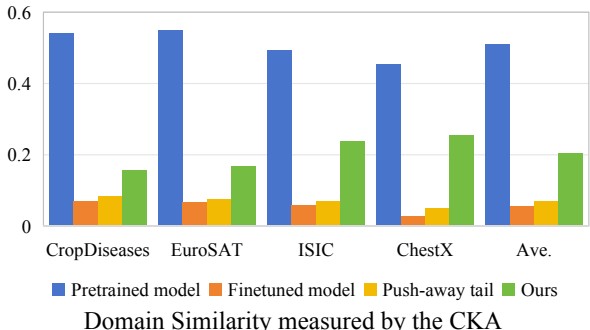

Domain Similarity measured by the CKA

*Figure 3.* Domain similarities by the CKA similarity, where lower similarity indicates more domain information. The abnormally low similarity validates that aligning tail tokens leads to overfitting.

---

[3]To adapt to target domains, the domain similarity should be lowered, but an abnormally low domain similarity may indicate overfitting to the target-domain few-shot training data.

information (head tokens); for tail tokens, breaking the alignment is more useful.

## 4. Method

Driven by our finding that aligning all tokens can be detrimental under domain shift, we propose the **Adaptive Tail-Head Alignment (ATHA)** method. Unlike the prevalent uniform alignment strategies, ATHA introduces an *asymmetric* alignment policy to simultaneously enhance the vision-text alignment of **Head Tokens** and suppress that of **Tail Tokens**, encouraging the model to learn useful information during vision-text alignment while resisting overfitting.

**Inputs and Feature Preparation.** Our method operates on the visual token sequence and class text embeddings within the CLIP framework. Given an input image $\mathbf{x} \in \mathbb{R}^{B \times C \times H \times W}$, it is first divided into $L$ patches and processed through the vision transformer blocks and outputs a sequence of visual tokens for each layer $l$: $\mathbf{V}^{(l)} = [\mathbf{v}_{[\text{CLS}]}^{(l)}; \mathbf{v}_1^{(l)}; \dots; \mathbf{v}_L^{(l)}] \in \mathbb{R}^{B \times (L+1) \times D}$, where we primarily work with the patch tokens $\mathbf{V}_{1:L}^{(l)}$. For the $N$ class names, we obtain their text embeddings $\mathbf{T} \in \mathbb{R}^{N \times D_t}$ from CLIP's text encoder and project them into the visual token space using CLIP's pretrained visual projection matrix $\mathbf{W}_p$:

$$\mathbf{T}' = \text{LayerNorm}(\mathbf{T})\mathbf{W}_p^\top \in \mathbb{R}^{N \times D}. \quad (3)$$

where $\mathbf{W}_p \in \mathbb{R}^{D \times D_t}$ is CLIP's visual projection matrix, $\mathbf{T}' \in \mathbb{R}^{K \times D}$ shares the same dimension as visual tokens, which is shared across all layers.

**Token-Text Similarity Calculation.** Given visual tokens $\mathbf{V}^{(l)} \in \mathbb{R}^{B \times L \times D}$ at layer $l$ and projected text embeddings $\mathbf{T}' \in \mathbb{R}^{N \times D}$ for $N$ classes, we compute the similarity:

$$s_{b,i,j}^{(l)} = \frac{\mathbf{v}_{b,i}^{(l)\top} \mathbf{t}_j'}{\|\mathbf{v}_{b,i}^{(l)}\|\|\mathbf{t}_j'\|}, \quad \forall b \in [1, B], i \in [1, L], j \in [1, N] \quad (4)$$

For each visual token $i$ in batch $b$, we find its maximum similarity to any class, $s_{b,i}^{\max,(l)} = \max_j s_{b,i,j}^{(l)}$, which serves as the proxy for its transferability.

**Discriminative Token Selection.** We select tokens for asymmetric processing based on $s_{b,i}^{\max,(l)}$:

(1) **Head Tokens** ($\mathcal{I}_{\text{head}}^{(l)}$): The top-$k_{\text{head}}$ tokens with the largest $s_{b,i}^{\max,(l)}$, where $k_{\text{head}} = \lfloor L \cdot \rho \rfloor$ and $\rho \in (0, 1)$ is the *head ratio*. These tokens likely encode transferable, class-discriminative patterns.

(2) **Tail Tokens** ($\mathcal{I}_{\text{tail}}^{(l)}$): The last-$r_{\text{tail}}$ tokens with the smallest $s_{b,i}^{\max,(l)}$, where $r_{\text{tail}} = \lfloor L \cdot \gamma \rfloor$ and $\gamma \in (0, 1)$ is the *tail ratio*. These tokens are suspected to carry domain-specific noise or non-transferable features.

Tokens not in either set remain unmodified.

**Asymmetric Token Alignment.** We introduce two layer-wise learnable parameters, $\alpha^{(l)}$ and $\beta^{(l)}$, to adaptively control the strength of pulling and pushing, respectively.

(1) For each **Head Token** at position $(b, i)$ where $i \in \mathcal{I}_{\text{head}}^{(l)}$, we identify its *most similar* text embedding $j^+ = \arg\max_j s_{b,i,j}^{(l)}$ and **enhance** it:

$$\tilde{\mathbf{v}}_{b,i}^{(l)} = \mathbf{v}_{b,i}^{(l)} + \alpha^{(l)} \cdot \mathbf{t}_{j^+}' \quad (5)$$

(2) For each **Tail Token** at position $(b, i)$ where $i \in \mathcal{I}_{\text{tail}}^{(l)}$, we identify its *least similar* text embedding $j^- = \arg\min_j s_{b,i,j}^{(l)}$ and **suppress** it:

$$\tilde{\mathbf{v}}_{b,i}^{(l)} = \mathbf{v}_{b,i}^{(l)} - \beta^{(l)} \cdot \mathbf{t}_{j^-}' \quad (6)$$

The complete token update at layer $l$ can be summarized as:

$$\tilde{\mathbf{v}}_{b,i}^{(l)} = \begin{cases} \mathbf{v}_{b,i}^{(l)} + \alpha^{(l)} \cdot \mathbf{t}_{j^+}', & \text{if } i \in \mathcal{I}_{\text{head}}^{(l)} \quad \text{(Pull)} \\ \mathbf{v}_{b,i}^{(l)} - \beta^{(l)} \cdot \mathbf{t}_{j^-}', & \text{if } i \in \mathcal{I}_{\text{tail}}^{(l)} \quad \text{(Push)} \\ \mathbf{v}_{b,i}^{(l)}, & \text{otherwise} \end{cases}$$

The modified tokens $\tilde{\mathbf{V}}^{(l)}$ are then passed through the remaining components of the transformer block:

$$\mathbf{V}^{(l+1)} = \text{TransformerBlock}^{(l)}(\tilde{\mathbf{V}}^{(l)}) \quad (7)$$

**Optimization.** The final classification is performed by computing similarity between the visual [CLS] token and text embeddings to compute the cross-entropy loss as Eq. 1.

The parameters $\{\alpha^{(l)}, \beta^{(l)}\}$ for every layer are optimized end-to-end with this standard cross-entropy loss, allowing the model to automatically learn the optimal intensity for strengthening head tokens and suppressing tail tokens per layer. This dynamic, discriminative alignment equips the model with a more robust mechanism for cross-domain few-shot finetuning than uniform alignment strategies.

## 5. Experiments

**Datasets and Evaluation Protocol.** Following the established benchmark for Cross-Domain Few-Shot Learning, we evaluate our method on four challenging target domains that exhibit significant distribution shifts from general source domains: CropDiseases (Mohanty et al., 2016), EuroSAT (Helber et al., 2019), ISIC2018 (Codella et al., 2019), and ChestX (Wang et al., 2017). We adopt the standard $N$-way $K$-shot evaluation protocol (with $N = 5$), conducting 800 episodes for 1-shot and 400 episodes for 5-shot settings to ensure statistical reliability.

**Implementation Details.** We use CLIP-ViT/B-16 (Radford et al., 2021) as our backbone. All experiments follow

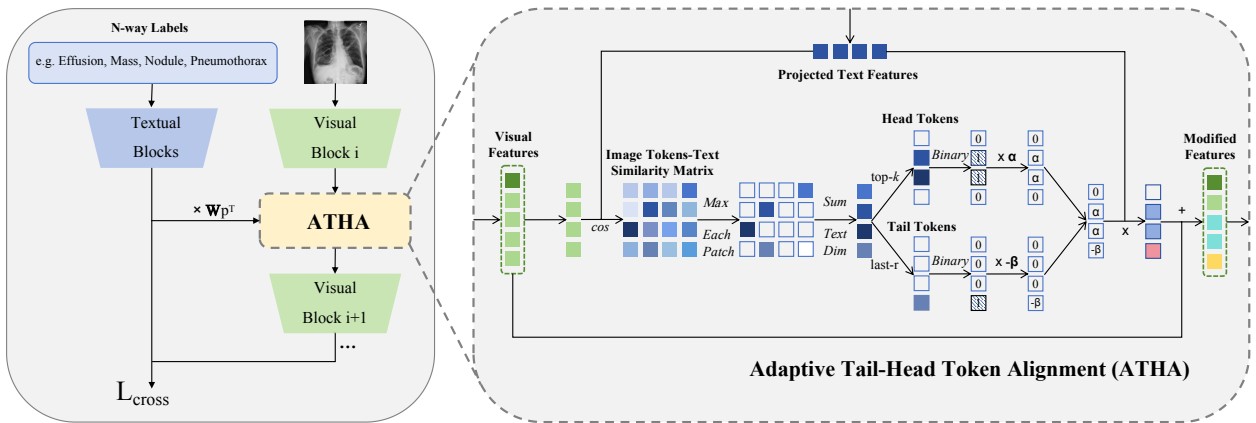

*Figure 4.* **Overview of our Adaptive Tail-Head Alignment (ATHA) framework.** Our method dynamically modulates visual tokens based on their semantic relevance to the target classes. At a given transformer layer, we compute the cosine similarity between each visual token and all class text embeddings. The top-$k_{\text{head}}$ tokens with the highest maximum similarity are identified as **Head Tokens** and are adaptively pulled closer to their most similar text embedding via a positive addition. Concurrently, the last-$r_{\text{tail}}$ tokens with the lowest maximum similarity are identified as **Tail Tokens** and are pushed away from their least similar text embedding via a negative subtraction. Layer-wise learnable parameters $\alpha^{(l)}$ and $\beta^{(l)}$ control the strength of these opposing operations.

the source-free setting, where only the pre-trained model and target-domain few-shot support set are accessible. Following the LoRA (Low-Rank Adaptation) fine-tuning strategy (Hu et al., 2022), we freeze the backbone parameters and only train low-rank adaptation matrices, ensuring parameter-efficient adaptation. During fine-tuning, we employ a cross-entropy loss $\mathcal{L}_{\text{cross}}$ and train for 100 epochs using the AdamW optimizer. Data augmentation includes standard random cropping and horizontal flipping.

The core of our approach is the Adaptive Tail-Head Alignment (ATHA) mechanism. For each target transformer block $l$ where ATHA is applied, we introduce two sets of layer-wise learnable parameters: $\alpha^{(l)}$ to control the strength of pulling head tokens closer, and $\beta^{(l)}$ to control the strength of pushing tail tokens away. We initialize these parameters strategically to focus the initial learning signal: $\alpha^{(8)} = 0.8$ and $\alpha^{(l)} = 0.0$ for other layers, $\beta^{(l)} = 0.01$ for all layers, allowing the model to first learn to enhance transferable features before adaptively learning to suppress noise. The token selection hyperparameters are set as $\rho = 0.1$ for the head token ratio and $\gamma = 0.1$ for the tail token ratio.

### 5.1. Comparison with State-of-the-Art Methods

We compare our method against several recent CDFSL approaches(Zanella & Ben Ayed, 2024; Xu et al., 2024a;b; Yazdanpanah & Moradi, 2022; Yi et al., 2025b;a; Zou et al., a;b; Ma et al., 2024; Zou et al., 2024; Fu et al., 2023) under both 5-way 1-shot and 5-shot settings in all target datasets. As shown in Tab. 1, our method achieves the best average performance across both 1-shot and 5-shot settings, outperforming all current state-of-the-art works. Notably, on the challenging ISIC2018 dataset, ATHA surpasses the

best previous method by 0.19% in 1-shot and 1.14% in 5-shot, demonstrating its robustness on domains with substantial distribution shifts. Furthermore, our method delivers the best or runner-up results on most individual datasets, with particularly strong gains on EuroSAT and CropDiseases. These results validate that our asymmetric alignment strategy effectively mitigates overfitting while preserving transferable features, leading to superior cross-domain fine-tuning performance.

### 5.2. Analysis of Tail Token Repulsion

A core claim of our method is that actively repelling tail tokens is essential for improving generalization in cross-domain few-shot learning. To validate this effect, we design a controlled experiment comparing three setups across all four benchmark datasets: (1) Baseline: standard fine-tuning without any token manipulation; (2) "Push-away tail" model: a strategy that applies only the "push-away" operation to identified tail tokens, without enhancing head tokens; (3) ATHA: which simultaneously repels tail tokens and enhances head tokens. The results, summarized in Tab. 2, reveal a critical finding.

The experimental results show that the Push-away tail strategy improves performance across all four datasets, demonstrating the general importance of suppressing tail tokens during cross-domain finetuning. This finding directly supports our core argument: under domain shift, forcing alignment of all tokens introduces overfitting to the training data, and selectively breaking the alignment of tail tokens can effectively enhance the model's learning in the cross-domain few-shot settings.

*Table 1.* Comparison with state-of-the-art works by the 5-way 1-shot and 5-way 5-shot classification.

| Method | Mark | Backbone | Shot | Source | Target | ISIC | EuroSAT | CropDiseases | ChestX | Ave. |
|---|---|---|---|---|---|---|---|---|---|---|
| StyleAdv-FT (Fu et al., 2023) | CVPR-23 | ViT/DINO | 1 | ✓ | ✓ | 33.99 | 74.93 | 84.11 | 22.92 | 53.99 |
| FLoR (Zou et al., 2024) | CVPR-24 | ViT/DINO | 1 | ✓ | ✓ | 35.49 | 73.09 | 83.55 | 23.26 | 53.85 |
| DAMIM (Ma et al., 2024) | AAAI-25 | ViT/DINO | 1 | ✓ | ✓ | 36.35 | 73.61 | 83.90 | 23.38 | 54.31 |
| CD-CLS (Zou et al., b) | NeurIPS-24 | ViT/DINO | 1 | ✓ | ✓ | 35.56 | 74.97 | 84.53 | 23.39 | 54.62 |
| AttnTemp (Zou et al., a) | NeurIPS-24 | ViT/DINO | 1 | ✓ | ✓ | 38.05 | 75.09 | 84.78 | 23.63 | 55.39 |
| ReCIT (Yi et al., 2025b) | ICML-25 | ViT/DINO | 1 | ✓ | ✓ | 38.48 | 75.23 | 85.92 | 23.84 | 55.87 |
| REAP (Yi et al., 2025a) | ICML-25 | ViT/DINO | 1 | ✓ | ✓ | 38.67 | 75.97 | 85.33 | **24.17** | 56.04 |
| FN+VDB (Yazdanpanah & Moradi, 2022) | CVPR-22 | RN18 | 1 | - | ✓ | 32.96 | 69.67 | 79.68 | 22.64 | 51.24 |
| IM-DCL (Xu et al., 2024b) | TIP-24 | RN10 | 1 | - | ✓ | 38.13 | 77.14 | 84.37 | 23.98 | 55.91 |
| StepSTP (Xu et al., 2024a) | TPAMI-25 | ViT/CLIP | 1 | - | ✓ | 32.97 | 70.01 | 84.84 | 22.84 | 52.68 |
| CLIP-LoRA (Zanella & Ben Ayed, 2024) | CVPRW-24 | ViT/CLIP | 1 | - | ✓ | 35.23 | 81.41 | 85.32 | 21.73 | 55.92 |
| **CLIP-LoRA + ATHA** | Ours | ViT/CLIP | 1 | - | ✓ | **38.86** | **82.56** | **87.99** | 24.00 | **58.35** |
| StyleAdv-FT (Fu et al., 2023) | CVPR-23 | ViT/DINO | 5 | ✓ | ✓ | 51.23 | 90.12 | 95.99 | 26.97 | 66.08 |
| FLoR (Zou et al., 2024) | CVPR-24 | ViT/DINO | 5 | ✓ | ✓ | 53.06 | 90.75 | 96.47 | 27.02 | 66.83 |
| DAMIM (Ma et al., 2024) | AAAI-25 | ViT/DINO | 5 | ✓ | ✓ | 54.86 | 91.18 | 96.34 | 27.82 | 67.55 |
| CD-CLS (Zou et al., b) | NeurIPS-24 | ViT/DINO | 5 | ✓ | ✓ | 54.69 | 91.53 | 96.27 | 27.66 | 67.54 |
| AttnTemp (Zou et al., a) | NeurIPS-24 | ViT/DINO | 5 | ✓ | ✓ | 54.91 | 90.82 | 96.66 | 28.03 | 67.61 |
| ReCIT (Yi et al., 2025b) | ICML-25 | ViT/DINO | 5 | ✓ | ✓ | 54.91 | 91.58 | 96.85 | 28.88 | 68.06 |
| REAP (Yi et al., 2025a) | ICML-25 | ViT/DINO | 5 | ✓ | ✓ | 55.28 | 91.79 | 96.71 | 28.34 | 68.03 |
| FN+VDB (Yazdanpanah & Moradi, 2022) | CVPR-22 | RN18 | 5 | - | ✓ | 47.48 | 87.31 | 94.63 | 25.55 | 64.74 |
| IM-DCL (Xu et al., 2024b) | TIP-24 | RN10 | 5 | - | ✓ | 52.74 | 89.47 | 95.73 | **28.93** | 66.72 |
| StepSTP (Xu et al., 2024a) | TPAMI-25 | ViT/CLIP | 5 | - | ✓ | 52.12 | 89.40 | 96.01 | 26.36 | 65.97 |
| CLIP-LoRA (Zanella & Ben Ayed, 2024) | CVPRW-24 | ViT/CLIP | 5 | - | ✓ | 51.10 | 92.52 | 96.21 | 24.13 | 65.99 |
| **CLIP-LoRA + ATHA** | Ours | ViT/CLIP | 5 | - | ✓ | **56.42** | **93.41** | **97.62** | 26.67 | **68.53** |

*Table 2.* Ablation study on ATHA under 5-way 5-shot setting.

| Method | CropDisease | EuroSAT | ISIC2018 | ChestX | Ave. |
|---|---|---|---|---|---|
| Baseline | 96.21 | 92.52 | 51.10 | 24.13 | 65.99 |
| + Push-away tail | 97.18 | 92.96 | 54.52 | 25.14 | 67.45 |
| **+ ATHA** | **97.62** | **93.41** | **56.42** | **26.67** | **68.53** |
| (a) Loss constraint | 96.19 | 92.64 | 53.68 | 24.53 | 66.76 |
| (b) Average texts alignment | 97.38 | 93.12 | 55.11 | 25.29 | 67.73 |
| (c) Fixed $\alpha, \beta$ | 97.49 | 93.26 | 56.16 | 26.12 | 68.26 |

*Table 3.* Comparison with existing alignment methods under 5-way 5-shot setting.

| Method | CropDisease | EuroSAT | ISIC2018 | ChestX | Ave. |
|---|---|---|---|---|---|
| Baseline | 96.21 | 92.52 | 51.10 | 24.13 | 65.99 |
| + PACL (Mukhoti et al., 2023) | 96.22 | 92.48 | 50.55 | 24.14 | 65.85 |
| + SPARC (Bica et al., 2024) | 95.69 | 91.14 | 51.21 | 24.17 | 65.55 |
| **+ ATHA** | **97.62** | **93.41** | **56.42** | **26.67** | **68.53** |

Our analysis further reveals that enhancing head tokens simultaneously with repelling tail tokens yields better performance. Our ATHA outperforms the "Push-away tail" model on all datasets. This indicates that while the "repulsion" operation alone can reduce overfitting, the ultimate performance ceiling relies on actively learning features. "Pushing away tail tokens" and "pulling closer head tokens" form a synergistic mechanism: the former "lightens the load" for the model by avoiding overfitting, while the latter "empowers" the model by explicitly strengthening its learning. This dynamic and discriminative recalibration of the token space is key to the robustness of our method in handling diverse domain shifts.

### 5.3. Analysis of How ATHA Works

We compute the token-text similarity distributions and CKA similarity of ATHA to analyze how it works. ATHA model achieves the most discriminative distribution in Fig. 2bcd. It combines and amplifies the desired effects: the alignment of tail tokens is strongly suppressed (left peak remains low), while the alignment of head tokens is dramatically enhanced. This demonstrates that ATHA successfully executes a dual strategy: it simultaneously pushes tail tokens away and pulls head tokens closer. The result in Fig. 3 shows a significant increase in CKA of ATHA compared to other models, validating its effectiveness against overfitting to the target-domain training data.

### 5.4. Comparison with Existing Alignment Methods

To validate the effectiveness of our work, we compare it with existing alignment methods. We list two representative alignment methods from our related work: PACL(Mukhoti et al., 2023) and SPARC(Bica et al., 2024). The results, presented in Tab.3, show that these methods not only underperform our approach but also yield an average performance lower than the baseline across four datasets. This comparison confirms that for few-shot fine-tuning under domain shift, our break alignment strategy is more effective than existing methods.

### 5.5. Analysis of Direct Feature Modulation

To validate our direct feature manipulation by adding or subtracting text embedding rather than conventional loss-based optimization, we design a ablation study comparing these two approaches of achieving discriminative alignment.

Distinct from ATHA, which adds text embedding to head tokens and subtracts that to tail tokens to modify the features

directly, the Loss constraint method uses explicit loss terms to achieve the same objective by $\mathcal{L}_{pull}$ and $\mathcal{L}_{push}$. One maximizes the similarity between head tokens and corresponding text embeddings, and another one minimizes the similarity between tail tokens and text embeddings. The total loss is $\mathcal{L} = \mathcal{L}_{cls} + \lambda_1 \mathcal{L}_{pull} + \lambda_2 \mathcal{L}_{push}$, where $\mathcal{L}_{cls}$ is the standard classification loss, and $\lambda_1$, $\lambda_2$ are balancing hyper parameters.

The results, summarized in Tab.2a, reveal that using the loss function to constrain shows only marginal improvements over the baseline (+0.8% on average across datasets), despite requiring careful tuning of two additional hyperparameters ($\lambda_1$, $\lambda_2$). This indicates that optimizing token alignment indirectly through loss functions is inefficient; the gradient signals must propagate through multiple layers and compete with the primary classification objective, diluting their intended effect.

We attribute this superiority to three factors: (1) Directness: Feature modulation operates immediately on the relevant representations without relying on gradient flow through deep networks; (2) Precision: We can exactly control which tokens are modified and by how much, whereas loss-based methods affect all parameters simultaneously; (3) Stability: Our method requires no additional hyperparameter tuning beyond the simple scaling factors $\alpha$ and $\beta$, which we find can be set consistently across domains.

This ablation study confirms that our approach of direct feature modulation for alignment is not merely an alternative implementation but a fundamentally more effective strategy. By explicitly and immediately adjusting token embeddings toward or away from textual concepts, we provide the model with clear, unambiguous signals about which visual patterns to emphasize or ignore, a capability that conventional loss-based training struggles to replicate efficiently.

### 5.6. Analysis of Selecting Text Strategies

We conduct a critical ablation study to examine the core mechanism of our text integration method. Specifically, we compare two strategies for selecting the textual guidance to modify image tokens: (1) using only the text embedding of the most similar class (MaxSim) and the least similar class (MinSim), and (2) using a weighted average of all class text embeddings (WeightedAvg). The comparison is performed under the 5-way 5-shot setting, and the results are summarized in Tab.2b.

Our method, which employs the MaxSim/MinSim strategy, demonstrates better performance than applying Average texts alignment(WeightedAvg strategy). For head tokens, MaxSim provides a stronger and clearer semantic pull towards a single, most-relevant class. Conversely, for tail tokens, subtracting the embedding of their least similar class

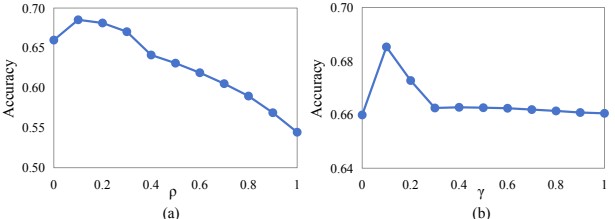

*Figure 5.* (a) Average performance sensitivity to the head token ratio $\rho$. The model achieves robust and competitive accuracy when $\rho$ falls within 0.1–0.3, with the peak observed at $\rho = 0.1$. This indicates that selecting a focused subset (10%–30%) of head tokens is optimal for adaptation. (b) Average performance sensitivity to the tail token ratio $\gamma$. Stable results are obtained when $\gamma$ is between 0.1 and 0.2, with the best performance at $\gamma = 0.1$. This confirms that suppressing only a small proportion (10%–20%) of tail tokens is sufficient for effective domain alignment.

creates a more decisive repulsive force, effectively enlarging the separation between tail patches and the most irrelevant class. The WeightedAvg strategy, which subtracts a blended textual vector, results in a weaker, less targeted repulsion.

This ablation validates the reasoning behind our design. The WeightedAvg strategy, while seemingly more nuanced, blurs the distinct semantic boundaries between classes. During finetuning, this can cause head tokens to absorb conflicting signals from multiple classes, and may not provide a strong enough directional signal to adequately push tail tokens away. Our MaxSim/MinSim strategy implements a clearer, more decisive rule: reinforce the strongest association, and repulse the weakest one, creating a sharper and more discriminative feature landscape, which is particularly crucial in the scarce data settings. The results confirm that a targeted, asymmetric alignment based on MaxSim/MinSim is more effective than a uniform blending of textual information for adapting CLIP under huge domain gaps.

### 5.7. Ablation Studies and Analysis

We conduct comprehensive ablation studies to validate parameters of our approach, using the 5-way 5-shot setting.

**Effect of Layer-wise Adaptive Scaling.** We evaluate the importance of our layer-specific learnable parameters $\alpha^{(l)}$ by comparing against fixed scaling strategies. Tab. 2c shows that using a single fixed scaling parameter for all layers leads to suboptimal performance, while our learnable layer-wise parameters achieve the best results. This demonstrates that different layers benefit from different alignment strengths.

**Sensitivity to Hyperparameters.** We investigate the sensitivity of our ATHA method to its core design hyperparameters: the head token selection ratio $\rho$ and the tail token selection ratio $\gamma$. As shown in Fig. 5, model performance remains robust across a wide range of $\rho$ and $\gamma$ values. Specifically, selecting 10%–30% of tokens as head tokens ($\rho \in [0.1, 0.3]$)

*Table 4.* Generalization across different backbones under 5-way 1-shot setting.

| Backbone | CropDisease | EuroSAT | ISIC2018 | ChestX | Ave. |
|---|---|---|---|---|---|
| CLIP (ViT-B/16) | 85.32 | 81.41 | 35.23 | 21.73 | 55.92 |
| + ATHA | **87.99** | **82.56** | **38.86** | **24.00** | **58.35** |
| CLIP (ViT-L/14) | 85.76 | 79.49 | 33.03 | 21.30 | 54.90 |
| + ATHA | **86.82** | **80.07** | **38.03** | **22.67** | **56.90** |
| SigLIP (Zhai et al., 2023) | 80.99 | 68.09 | 28.90 | 21.33 | 49.83 |
| + ATHA | **84.80** | **71.98** | **32.67** | **22.97** | **53.11** |
| PEcore(Bolya et al., 2026) | 87.14 | 77.80 | 39.34 | 21.97 | 56.61 |
| + ATHA | **89.81** | **80.33** | **40.52** | **22.48** | **58.29** |

and 5%–20% as tail tokens ($\gamma \in [0.05, 0.2]$) yields stable and competitive results, with the optimal configuration observed at $\rho = 0.1$ and $\gamma = 0.1$.

## 5.8. Generalization across Different Backbones

To demonstrate that our ATHA mechanism is not tied to a specific visual encoder, we evaluate its effectiveness on four different backbones under the 5-way 1-shot setting: CLIP (ViT-B/16) (Radford et al., 2021), CLIP (ViT-L/14), SigLIP (Zhai et al., 2023), and PEcore (Bolya et al., 2026). All backbones are kept frozen and adapted via LoRA fine-tuning with or without our ATHA module. As shown in Tab. 4, ATHA consistently improves performance across all architectures and datasets, yielding an average gain of +2.43%, +2.00%, +3.28%, and +1.68% respectively. This strong backbone-agnostic generalizability confirms that our ATHA strategy captures a universal cross-domain adaptation principle, independent of the specific pre-training objective or architecture.

## 5.9. Generalization across Fine-Tuning Frameworks

Our ATHA module can be combined with various parameter-efficient fine-tuning strategies. We integrate ATHA into three representative frameworks: CoOp (Zhou et al., 2022), MaPLe (Khattak et al., 2023), and LoRA (Hu et al., 2022), and evaluate them under the 5-way 5-shot setting. Tab. 5 shows that ATHA brings consistent and substantial improvements over the base methods, with average gains of +1.74%, +2.06%, and +2.54% respectively. Notably, the absolute improvement on the challenging ISIC2018 dataset reaches up to +5.32%. These results highlight the flexibility and broad applicability of our approach: regardless of whether the fine-tuning method learns input prompts (CoOp), deep prompt tokens (MaPLe), or low-rank weight updates (LoRA), ATHA's token-level repulsion-and-attraction mechanism provides an orthogonal and complementary benefit.

## 5.10. Choice of Similarity Metric

The core of ATHA relies on measuring the similarity between image patch tokens and class text embeddings to select head and tail tokens. While CLIP is pre-trained with cosine similarity, other distance or dependency measures could

*Table 5.* Generalization across different fine-tuning frameworks under 5-way 5-shot setting.

| Fine-tuning Strategy | CropDisease | EuroSAT | ISIC2018 | ChestX | Ave. |
|---|---|---|---|---|---|
| CoOp (Zhou et al., 2022) | 91.88 | 83.22 | 43.36 | 22.69 | 60.29 |
| + ATHA | **93.33** | **83.81** | **46.67** | **24.32** | **62.03** |
| MaPLe (Khattak et al., 2023) | 96.22 | 90.80 | 50.97 | 24.12 | 65.53 |
| + ATHA | **96.93** | **92.02** | **55.91** | **25.49** | **67.59** |
| LoRA (Hu et al., 2022) | 96.21 | 92.52 | 51.10 | 24.13 | 65.99 |
| + ATHA | **97.62** | **93.41** | **56.42** | **26.67** | **68.53** |

*Table 6.* Comparison of different similarity metrics for token selection under 5-way 5-shot setting.

| Distance Metric | CropDisease | EuroSAT | ISIC2018 | ChestX | Ave. |
|---|---|---|---|---|---|
| Baseline | 96.21 | 92.52 | 51.10 | 24.13 | 65.99 |
| Euclidean distance | 97.55 | 93.42 | 56.27 | 26.31 | 68.39 |
| Learned metric | 97.41 | 93.22 | **56.59** | 26.61 | 68.46 |
| Mutual information | 97.22 | **93.44** | 55.92 | **27.04** | 68.41 |
| **Cosine similarity** | **97.62** | 93.41 | 56.42 | 26.67 | **68.53** |

also be used. We compare cosine similarity (Chen et al., 2020) against three alternatives: Euclidean distance (Vinyals et al., 2016), a learned bilinear metric (Sung et al., 2018) (trained on the support set), and mutual information (Belghazi et al., 2018) estimated via kernel density. As shown in Tab. 6, all metrics achieve comparable performance, with mutual information even giving a slightly higher result on ChestX. This demonstrates that ATHA is not sensitive to the specific choice of similarity measure; the core benefit stems from the selective repulsion and attraction mechanism itself. Nevertheless, cosine similarity yields the best average performance (68.53%) and aligns naturally with CLIP's pre-training, so we adopt it as the default.

## 6. Conclusion

This paper identifies a counterintuitive phenomenon in target-domain finetuning of CDFSL: actively pushing away certain image tokens from text embeddings improves target-domain performance, challenging the prevailing uniform alignment paradigm. We demonstrate that the common belief of alignment is valid only for tokens already containing sufficient semantic information (head tokens); for tail tokens, breaking the alignment is more useful. Based on this insight, we propose Adaptive Tail-Head Alignment (ATHA), which dynamically strengthens or suppresses token alignment via learnable scaling parameters to control adding or subtracting text embedding for effective manipulation of alignment. Extensive experiments validate our state-of-the-art performance across multiple benchmarks. Future work will extend ATHA to other vision-language models beyond CLIP and explore its applicability to broader transfer learning scenarios, including domain generalization and few-shot class-incremental learning. Moreover, investigating adaptive token selection strategies that dynamically adjust $\rho$ and $\gamma$ per episode could further enhance robustness across diverse domain shifts.

## Acknowledgments

This work is supported by the National Natural Science Foundation of China under grants 62206102; the National Key Research and Development Program of China under grant 2024YFC3307900; the National Natural Science Foundation of China under grants 62436003, 62376103 and 62302184; Major Science and Technology Project of Hubei Province under grant 2025BAB011 and 2024BAA008; Hubei Science and Technology Talent Service Project under grant 2024DJC078; and Ant Group through CCF-Ant Research Fund. The computation is completed in the HPC Platform of Huazhong University of Science and Technology.

## Impact Statement

We introduce a CD-FSL method that dynamically strengthens or suppresses token alignment via learnable scaling parameters to control adding or subtracting text embedding for effective manipulation of alignment in the CLIP. This helps mitigate the domain gap and improves generalization to the target domain. Our approach is also applicable to other areas, such as domain generalization, domain adaptation, and few-shot class-incremental learning, where improving model transferability is a common challenge. While our evaluations focus on four distinct target domains, these may not encompass all potential real-world scenarios. Therefore, further evaluation across a wider range of target domains is needed to validate the approach in more realistic settings.

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

# Appendix for Improving CLIP Adaptation by Breaking Tail Alignment for Source-Free Cross-Domain Few-Shot Learning

## A. Datasets

We evaluate our method on four widely-used CDFSL benchmarks that exhibit substantial domain shifts from the source domain of natural images.

**CropDiseases** (Mohanty et al., 2016) is an agricultural dataset designed for plant disease classification, containing 43,456 high-resolution images across 38 disease categories. The images focus on close-up views of plant leaves, introducing a significant domain gap through their specialized content and detailed textures.

**EuroSAT** (Helber et al., 2019) is a remote sensing dataset for land use and land cover classification, consisting of 27,000 satellite images divided into 10 classes. The aerial perspective and unique spectral characteristics of satellite imagery create a distinct visual domain compared to conventional photographs.

**ISIC2018** (Codella et al., 2019) is a medical dataset comprising 10,015 dermoscopic images for skin lesion analysis across 7 categories. The clinical imaging conditions and specialized visual patterns of skin lesions represent a substantial domain shift from everyday images.

**ChestX** (Wang et al., 2017) is a medical radiology dataset containing 25,847 chest X-ray images spanning 7 thoracic conditions. The monochromatic modality, anatomical focus, and absence of natural scene elements result in the most pronounced domain gap among the evaluated datasets.

These four datasets, spanning agriculture, remote sensing, dermatology, and radiology, provide diverse and challenging target domains with progressively increasing domain shifts from the source domain.

## B. More similarity distribution

All target domain similarity distribution in Fig.6 indicate that inhibiting the alignment of tail tokens contributes majorly to the performance gain.

## C. Centered Kernel Alignment (CKA) Methodology

We employ Centered Kernel Alignment (CKA) (Kornblith et al., 2019) to quantitatively measure the similarity between feature representations across different domains. CKA is a

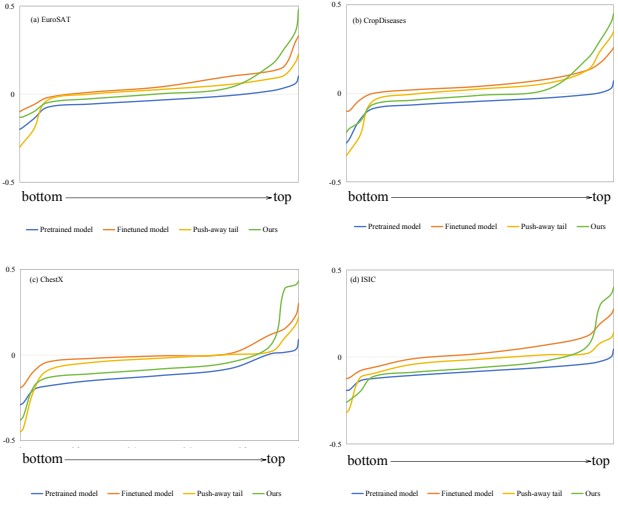

*Figure 6.* On the target four domains, the pretrained model (transferred directly) shows a flat, less discriminative curve due to domain shift. Standard fine-tuning causes a global upward shift, pulling both head and tail tokens closer to text. The Push-away-tail method suppresses the alignment of tail tokens while still enhancing head tokens, indicating that inhibiting the alignment of tail tokens contributes majorly to the performance gain.

widely adopted metric for comparing neural network representations, particularly effective in analyzing cross-domain relationships.

Given two sets of feature representations $X \in \mathbb{R}^{n \times d}$ and $Y \in \mathbb{R}^{n \times d}$, we first compute their Gram matrices:

$$K = XX^\top, \quad L = YY^\top \qquad (8)$$

which capture the pairwise similarity structure within each representation set. To eliminate the influence of feature means, we center these Gram matrices using:

$$K_c = HKH \qquad (9)$$
$$L_c = HLH \qquad (10)$$

where $H = I_n - \frac{1}{n}\mathbf{1}_n\mathbf{1}_n^\top$ is the centering matrix, with $I_n$ being the identity matrix and $\mathbf{1}_n$ a vector of ones.

The CKA similarity is then computed as the normalized inner product between the vectorized centered Gram matrices:

$$\mathrm{CKA}(X,Y) = \frac{\langle \mathrm{vec}(K_c), \mathrm{vec}(L_c)\rangle}{\|\mathrm{vec}(K_c)\|\|\mathrm{vec}(L_c)\|} = \frac{\mathrm{Tr}(K_c L_c)}{\sqrt{\mathrm{Tr}(K_c^2)\mathrm{Tr}(L_c^2)}} \qquad (11)$$

yielding values between 0 (completely dissimilar) and 1 (identical relational structures).

In our analysis, CKA serves to quantify the domain distance between source and target representations. Higher CKA values indicate greater domain similarity and less domain-specific information in the features, while lower values suggest stronger domain shifts and more domain-characteristic representations. This metric provides crucial insights into the model's generalization behavior and adaptation characteristics across diverse data distributions.

