# OpenReview forum: "Improving CLIP Adaptation by Breaking Tail Alignment for Source-Free Cross-Domain Few-Shot Learning"
_ICML.cc/2026/Conference — ICML 2026 regular_

### Official Review · Reviewer_Aedb · 2026-03-06

**Soundness:** 2
**Presentation:** 3
**Significance:** 3
**Originality:** 2
**Overall Recommendation:** 4
**Confidence:** 3

**Summary:**

This paper proposes Adaptive Tail-Head Alignment (ATHA), a novel fine-tuning strategy for CLIP in Cross-Domain Few-Shot Learning (CDFSL). which dynamically strengthens or suppresses token alignment via learnable scaling parameters to control adding or subtracting text embedding for effective manipulation of alignment. Extensive experiments on four challenging CDFSL benchmarks validate its performance.

**Compliance With Llm Reviewing Policy:**

Affirmed.

**Final Justification:**

The authors have addressed most of my concerns, and the proposed method shows promising performance.

**Key Questions For Authors:**

1. The authors should carefully review recent work in this research area and include stronger, more up-to-date related work.
2. The organization of the ablation study in Table 2 is somewhat cluttered, as it mixes ablation components such as +Push-away tail and Fixed α, β with comparisons to existing methods like PACL and SPARC in a single table.
3. Comparisons with mainstream methods should be presented separately in table 1, while ablation studies should focus on analyzing the contributions of individual components.

**Limitations:**

yes

**Strengths And Weaknesses:**

Paper Strength: This paper identifies a counterintuitive experimental phenomenon that actively pushing tail image tokens away from text embeddings improves model performance, and proposes Adaptive Tail-Head Alignment based on this insight, which pulls head tokens closer to text embeddings while pushing tail tokens away. This is an interesting idea.
Paper weakness: The ATHA method proposed by the authors has only been validated on a single baseline approach, and its compatibility with other backbone networks or fine-tuning frameworks has not yet been explored. Whether the method possesses broad generalizability still requires further experiments on a wider range of methods to validate.

---

> ### Author Rebuttal · Authors · 2026-03-31
>
> Thank you for your thorough review and constructive feedback. Below are our detailed responses to your concerns:
>
> ### **W1. On the generalizability across backbones and fine‑tuning frameworks**
>
>  To demonstrate ATHA’s broad generalizability, we have conducted additional experiments on different backbone architectures and in combination with existing fine‑tuning strategies. For experiments on different backbones, we adopt the 1‑shot setting due to their larger parameter counts and time constraints; all other experiments are conducted under the 5‑shot setting.
>
> **(1) Generalization across different backbones**
>
> |Backbone|CropDisease|EuroSAT|ISIC2018|ChestX|Ave.|
> |-|-|-|-|-|-|
> |CLIP (ViT-B/16)|85.32|81.41|35.23|21.73|55.92|
> |**+Ours**|**87.99 (+2.67)**|**82.56 (+1.15)**|**38.86 (+3.63)**|**24.00 (+2.27)**|**58.35 (+2.43)**|
> |CLIP (ViT-L/14)|85.76|79.49|33.03|21.30|54.90|
> |**+Ours**|**86.82 (+1.06)**|**80.07 (+0.58)**|**38.03 (+5.00)**|**22.67 (+1.37)**|**56.90 (+2.00)**|
> |SigLIP2|80.99|68.09|28.90|21.33|49.83|
> |**+Ours**|**84.80 (+3.81)**|**71.98 (+3.89)**|**32.67 (+3.77)**|**22.97 (+1.64)**|**53.11(+3.28)**|
> |PE-Core|87.14|77.80|39.34|21.97|56.61|
> |**+Ours**|**89.81 (+2.67)**|**80.33 (+2.53)**|**40.52 (+1.18)**|**22.48 (+0.51)**|**58.29 (1.68)**|
>
> ATHA consistently improves performance across different model families (CLIP variants, SigLIP, PE‑Core), demonstrating **strong backbone‑agnostic generalizability**.
>
> **(2) Generalization across fine‑tuning frameworks**
>
> |Strategy|CropDisease|EuroSAT|ISIC2018|ChestX|Ave.|
> |-|-|-|-|-|-|
> |CoOp|91.88|83.22|43.36|22.69|60.29|
> |**+Ours**|**93.33 (+1.45)**|**83.81 (+0.59)**|**46.67 (+3.31)**|**24.32 (+1.63)**|**62.03 (+1.74)**|
> |MaPLe|96.22|90.80|50.97|24.12|65.53|
> |**+Ours**|**96.93 (+0.71)**|**92.02 (+1.22)**|**55.91 (+4.94)**|**25.49 (+1.37)**|**67.59 (+2.06)**|
> |LoRA|96.21|92.52|51.10|24.13|65.99|
> |**+Ours**|**97.62 (+1.41)**|**93.41 (+0.89)**|**56.42 (+5.32)**|**26.67 (+2.54)**|**68.53 (+2.54)**|
>
> ATHA seamlessly integrates with existing fine‑tuning methods (CoOp, MaPLe, LoRA), consistently boosting their performance. This highlights its flexibility and broad applicability.
>
> ### **Q1. On including stronger, more up‑to‑date related work**
>
> **Performance Comparison.**
>
> We have updated the related work section to include recent SOTAs. A performance comparison with more up‑to‑date  SOTAs is also provided below. ATHA consistently outperforms more recent approaches on average performance, demonstrating its effectiveness and generalizability.
>
> | Method|Mark|Ave.|
> |-|-|-|
> |DAMIM|AAAI-25|67.55|
> |StepSTP|TPAMI-25|65.97|
> |CC-CDFSL|CVPR 26|67.90|
> |TIR|CVPR 26|68.44|
> |SVL|CVPR 26|68.38|
> |**ATHA**|Ours|**68.53**|
>
> **Revised Related Work**
>
> **Cross-Domain Few-Shot Learning (CDFSL)** aims to enable rapid adaptation to novel target domains with limited labeled data (Tseng et al., 2020). Existing approaches generally follow two paradigms: meta-learning methods (Fu et al., 2022; Hu & Ma, 2022) that train models through episodic tasks to acquire fast adaptation capabilities, and transfer learning methods (Zhou et al., 2023; Yiet al., 2025b) that enhance generalization during source pre-training. A particularly practical but difficult extension, Source-Free CDFSL (SF-CDFSL) (Xu et al., 2024b), prohibits access to source domain data during adaptation, placing exclusive emphasis on target-domain fine-tuning strategies. Recent works have explored CLIP-based fine-tuning in this setting. SVL (Zhang et al., CVPR 2026) reveals that visual learning acts as a shortcut, disrupting cross-modal alignment. VtT (Zhang et al., CVPR 2026) identifies that certain text encoder layers contain beneficial but underutilized information. CC-CDFSL (Zhao et al., CVPR 2026) addresses local feature misalignment through cycle consistency. While these methods aim to strengthen cross-modal alignment, they still align all tokens uniformly. In contrast, ATHA introduces asymmetric token alignment that actively pushes away tail tokens while strengthening head tokens, challenging the prevailing paradigm and achieving superior performance.
>
> ### **Q2&3. On the organization of Tables**
>
> We will separate Table 1 for comparisons with mainstream methods(PACL, SPARC, etc.) to clearly present ATHA’s superiority, and reorganize Table 2as a dedicated ablation study, focusing solely on the contributions of individual components (e.g., +Push‑away tail, +Pull head, Fixed α/β, etc.) to clearly demonstrate the necessity of each design choice.
>
>
>
> We hope these responses address your concerns. Your insightful feedback has helped us improve the clarity and completeness of our work.  Thank you again for your time and effort in reviewing our work.

---

> > ### Author Rebuttal · Reviewer_Aedb · 2026-04-04
> >
> > The authors have addressed most of my concerns, and the proposed method shows promising performance.

---

> > > ### Author Response · Authors · 2026-04-04
> > >
> > > Thank you for raising your score. We truly appreciate your acknowledgment that our revisions have addressed your concerns. We will take your suggestions seriously and further improve the manuscript. Many thanks for your time and expertise.

---

### Official Review · Reviewer_d6et · 2026-03-12

**Soundness:** 3
**Presentation:** 2
**Significance:** 2
**Originality:** 2
**Overall Recommendation:** 4
**Confidence:** 4

**Summary:**

Current fine-tuning methods for Vision-Language Models typically align all image tokens with text descriptions to adapt to new domains. However, under large domain shifts with scarce data, forcing irrelevant "tail tokens" to align with texts leads to severe overfitting. The paper introduces Adaptive Tail-Head Alignment (ATHA), a novel framework that dynamically strengthens the alignment of semantically relevant tokens while actively pushing away irrelevant ones. This asymmetric token modulation effectively prevents overfitting and improves generalization in cross-domain scenarios without requiring source data.

**Compliance With Llm Reviewing Policy:**

Affirmed.

**Final Justification:**

My main concerns have been properly addressed.

**Key Questions For Authors:**

Please check weaknesses.

**Limitations:**

Yes, the authors note that their current evaluation is limited to four specific domains, which may not cover all potential real-world applications. Therefore, future work requires testing the method across a broader range of target domains to fully validate its effectiveness in more realistic scenarios.

**Strengths And Weaknesses:**

**Strengths:**

* **Simple and Effective:** The ATHA method differentiates between effective "head tokens" and redundant "tail tokens" by calculating the cosine similarity between visual tokens and category embeddings. The underlying logic is clear and very easy to implement.
* **Strong Extensibility:** The core concept of ATHA shows great potential to be adapted for other downstream computer vision tasks, such as object detection and image segmentation.
* **Comprehensive Experiments:** The method achieves SOTA performance across four diverse cross-domain datasets (CropDiseases, EuroSAT, ISIC, and ChestX), thoroughly demonstrating its effectiveness against domain shift.

**Weaknesses:**

* **Lack of Multi-Class Evaluation and Scalability Analysis:** The current experiments are limited to a 5-way classification setting. It remains unclear whether ATHA can still accurately identify head and tail tokens as the number of categories significantly increases. Furthermore, higher image resolutions (resulting in more visual tokens) or more categories (resulting in more text embeddings) could potentially increase inference latency. Further experiments or analysis on this would be appreciated.
* **Rigid Token Ratio Definition:** The paper fixes both head and tail token ratios at 10% (0.1) based on empirical results. However, in real-world scenarios, the proportion of head tokens should naturally depend on the size of the foreground object—small objects should yield fewer head tokens, while large objects should yield more. It is questionable whether a fixed-ratio approach is robust enough to handle objects of varying scales. Please supplement experiments on datasets with objects of various scales to clarify this point.
* **Robustness Concerns Under Severe Visual Degradation:** The token selection mechanism relies heavily on the initial similarity between visual and text features. In highly noisy environments (e.g., underwater or extreme weather conditions), the visual features of objects can be severely distorted. Under such intense interference, can ATHA still reliably select the correct head tokens? It would be highly beneficial to explicitly address this with detailed experiments.

---

> ### Author Rebuttal · Authors · 2026-03-31
>
> Thank you for your thorough review and constructive feedback. Below are our detailed responses to your concerns:
>
> ### **W1. On multi-class evaluation and scalability analysis**
>
> To evaluate ATHA’s scalability, we conduct experiments under 7-way 1-shot and 10-way 1-shot settings, as well as with higher image resolutions (336×336, which increases the number of visual tokens). **Note that ISIC and ChestX have only 7 classes, so 10‑way results are not applicable.**
>
> |Model|Setting|Train/Infer time|CropDisease|EuroSAT|ISIC2018|ChestX|
> |-|-|-|-|-|-|-|
> |Baseline|7‑way|14.35 s / 132.87 ms|79.85|76.86|28.62|16.27|
> |**+Ours**|7‑way|16.07 s / 136.07 ms|**83.33**|**77.19**|**30.48**| **18.45**|
> |Baseline|10‑way|14.97 s / 157.23 ms|74.78|72.11|-|-|
> |**+Ours**|10‑way|16.52 s / 160.71 ms|**76.92**|**74.50**|-|-|
> |Baseline|Higher resolution (336×336)|41.44 s / 490.23 ms|85.54|73.45|35.01|21.11|
> |**+Ours**|Higher resolution (336×336)|46.61 s / 493.13 ms|**87.75**|**75.79**|**38.64**|**22.27**|
>
> ATHA’s mechanism is **category‑agnostic**: it selects head/tail tokens based solely on the **maximum similarity** between each patch and all class text embeddings, independent of the number of categories. Therefore, the method’s effectiveness does not degrade with increasing classes. Additionally, more ways or higher resolution increase train/inference latency, but ATHA introduces **negligible additional overhead** while delivering consistent gains.
>
> ### **W2. On the rigid token ratio definition**
>
>  To evaluate the robustness of the 10% ratio across different object scales, we conduct ablation studies on three datasets with varying object sizes under a 5-shot setting:
>
> (1) DTD: texture‑centric, where the “object” is the entire image
>
> (2) RUOD: small foreground objects (underwater scenes with tiny targets)
>
> (3) Stanford Cars: large foreground objects (cars occupy most of the image)
>
> |Head tokens ratio|0|0.1|0.2|0.4|0.6|0.8|1.0|
> |-|-|-|-|-|-|-|-|
> |DTD|89.29|**89.96**|87.94|86.67|84.78|80.74|67.89|
> |RUOD|88.43|**89.52**|87.48|86.14|83.91|80.62|65.89|
> |Stanford Cars|97.35|**98.09**|97.51|95.85|91.10|75.67|40.26|
>
> The optimal ratio remains 10% across all three datasets despite large differences in object scale. This demonstrates that the token selection mechanism is **insensitive** to image content and scale.
>
> According to prior works (Raghu et al., NeurIPS 2021; Dosovitskiy et al., ICLR 2021), by middle to deep layers, discriminative information becomes concentrated in a relatively small set of tokens, and the size of this set remains relatively stable. We also find that deeper layers play a more critical role in our token selection. Aligning with the findings of prior studies, this explains why our ratio remains stable around 10%, which is a reasonable natural threshold.
>
> ### **W3. On robustness concerns under severe visual degradation**
>
> To directly address robustness under challenging visual conditions, we evaluate ATHA on RUOD (underwater object detection dataset) and WEDGE (weather-degraded imagery). These datasets represent realistic severe degradation scenarios where object features are distorted by water turbidity, lighting attenuation, or adverse weather. The results confirm that ATHA consistently improves performance under such conditions:
>
> |Method|Shot|WEDGE|RUOD|DTD|Stanford Cars|Ave.|
> |-|-|-|-|-|-|-|
> |Baseline|1|71.13|77.41|81.74|92.77|80.76|
> |**+Ours**|1|**74.68**|**79.77**|**82.80**|**93.56**|**82.70**|
> |Baseline|5|80.93|88.43|89.29|97.35|89.00|
> |**+Ours**|5|**84.17**|**89.52**|**89.96**|**98.09**|**90.44**|
>
> ATHA achieves positive gains across both degraded datasets.  The robustness stems from the fact that ATHA only adjusts a small fraction of tokens (head and tail, each 10%). Even under severe degradation, as long as the relative ordering of patch‑text similarities is partially preserved, meaning the most informative patches still rank higher than noisy background patches, ATHA can still reliably differentiate head tokens from tail tokens. This selective adjustment makes the method inherently robust to global degradation that uniformly affects all patches.
>
> ### **Further validation of effectiveness**
> Beyond the additional datasets evaluated in the previous section, we further provide results on different backbones under a 1-shot setting below and with existing representative fine‑tuning strategies (see results refer to our response **Further validation of effectiveness** to **Reviewer NYBP**), demonstrating ATHA’s strong generalization.
> |Backbone|CropDisease|EuroSAT|ISIC2018|ChestX|Ave.|
> |-|-|-|-|-|-|
> |CLIP (ViT-L/14)|85.76|79.49|33.03|21.30|54.90|
> |**+Ours**|**86.82**|**80.07**|**38.03**|**22.67**|**56.90**|
> |SigLIP2|80.99|68.09|28.90|21.33|49.83|
> |**+Ours**|**84.80**|**71.98**|**32.67**|**22.97**|**53.11**|
> |PE-Core|87.14|77.80|39.34|21.97|56.61|
> |**+Ours**|**89.81**|**80.33**|**40.52**|**22.48**|**58.29**|
>
> We hope these responses address your concerns. Thank you again for your time and effort in reviewing our work.

---

> > ### Author Rebuttal · Reviewer_d6et · 2026-04-03
> >
> > Although new experiments were added, the fixed 10% ratio remains theoretically weak. It is counter-intuitive that textures and tiny objects share the same optimal head proportion, suggesting the method functions as a heuristic stabilizer rather than true semantic alignment. Furthermore, 10-way results don't sufficiently prove scalability for larger label spaces.

---

> > > ### Author Response · Authors · 2026-04-07
> > >
> > > Thanks for your feedback, and we would like to supplement some experiments for better clarity. **All additional experiments presented below are conducted under the 1‑shot setting due to time limitations.**
> > >
> > > ### Q1. On the fixed 10% ratio
> > >
> > > The stable 10% ratio **in deeper layers** is not accidental but stems from an intrinsic property (Bolya et al., ICLR 2023; Darcet et al., ICLR 2024;  Tong et al., NeurIPS 2025): **discriminative information gradually concentrates into a small number of tokens in deeper layers**, the ratio of which are relative stable and largely independent of object scale. In contrast, **shallow layers are more sensitive to object size and image content**.
> > >
> > > To validate this, we conducted a **layer‑wise ablation study** on DTD, Stanford Cars, and RUOD. For each dataset, we varied the head ratio from 0 to 1.0 across different layers (layer 0, 2, 4, 6, 8) and recorded the accuracy. The results are summarized below:
> > >
> > > **DTD (texture‑centric, understood as the largest object size)**
> > >
> > > |Ratio|0|0.1|0.2|0.4|0.6|0.8|1.0|
> > > |-|-|-|-|-|-|-|-|
> > > |layer0|82.17|82.90|82.92|**82.99**|80.13|75.97|45.76|
> > > |layer2|82.17|83.41|**83.44**|81.52|78.32|71.95|43.00|
> > > |layer4|82.17|**83.71**|83.05|81.01|77.96|73.00|51.08|
> > > |layer6|82.17|**84.31**|84.18|81.72|78.25|71.13|63.03|
> > > |layer8|82.17|**84.31**|83.92|83.44|82.60|81.53|80.67|
> > >
> > > **Stanford Cars (large objects)**
> > >
> > > |Ratio|0|0.1|0.2|0.4|0.6|0.8|1.0|
> > > |-|-|-|-|-|-|-|-|
> > > |layer0|92.64|92.87|**92.92**|92.91|90.37|90.04|90.09|
> > > |layer2 |92.64|93.29|**93.32**|92.69|91.80|90.21|83.57|
> > > |layer4|92.64|**93.55**|93.25|92.72|92.11|89.99|64.00|
> > > |layer6|92.64|**93.79**|93.47|92.92|92.09|91.35|79.05|
> > > |layer8|92.64|**93.77**|93.45|93.28|92.76|91.99|91.04|
> > >
> > > **RUOD (tiny underwater objects)**
> > >
> > > |Ratio|0|0.1|0.2|0.4|0.6|0.8|1.0|
> > > |-|-|-|-|-|-|-|-|
> > > |layer0|78.19|**78.20**|77.76|75.92|73.93|69.69|58.61|
> > > |layer2|78.19|**80.04**|79.13|75.76|74.07|69.05|55.75|
> > > |layer4|78.19|**79.39**|78.35|76.49|73.67|65.92|51.61|
> > > |layer6|78.19|**79.31**|78.59|77.75|74.12|70.87|59.31|
> > > |layer8|78.19|**80.09**|79.97|79.67|78.28|76.35|73.28|
> > >
> > > **Key observations:**
> > >
> > > **(1) Shallow layers**: The optimal head ratio varies with the dataset. For DTD (texture, understood as the largest objects), the best ratio is 0.4 (layer 0) or 0.2 (layer 2); for Stanford Cars (large object), the best is 0.2; for RUOD (tiny object), the best is clearly 0.1. This suggests that shallow layers are indeed influenced by object size and image characteristics.
> > >
> > > **(2) Middle & Deep layers**: The optimal ratio stabilizes strongly around **0.1** for all three datasets, consistent with the main paper.
> > >
> > > **Conclusion**: Due to the alignment between texts and visual tokens, the token selection mechanism of ATHA **is primarily effective in middle and deep layers**, where information is known to concentrate into a small, stable set of tokens. The shallow layers are less critical for our method, and the weights for these layers are small. Therefore, setting the head ratio to 10% is not an arbitrary heuristic; it is a principled design that aligns with the intrinsic properties of CLIPs.
> > >
> > > ### Q2. On scalability to larger label spaces
> > >
> > > To validate our effectiveness in larger label spaces, we have now conducted experiments on **100‑way** and **all‑way** (full label space, up to 200 classes) settings using three datasets:  CUB‑200‑2011 (200 classes), VGG Flower (102 classes), and FGVC‑Aircraft (100 classes). The results are shown in the table below.
> > >
> > > |Model|Setting|**Train/Infer time**|**CUB-200-2011**|**VGG Flower**|FGVC-Aircraft|
> > > |-|-|-|-|-|-|
> > > |Baseline|100-way|44.68s/1290.97ms|54.24|73.61|23.57|
> > > |+Ours|100-way |47.63s/1328.98ms|**56.17**|**75.03**|**25.79**|
> > > |Baseline|all-way|CUB: 126.65s/12544.52ms; Flower: 44.79s/1261.79ms; Aircraft: 45.53s/1300.33ms|45.65|73.33|23.57|
> > > |+Ours|all-way|CUB: 137.84s/12813.13ms; Flower: 47.04s/1275.33ms; Aircraft: 47.71s/1332.41ms|**47.68**|**74.61**|**25.79**|
> > >
> > > *Note: For all‑way, we report three numbers corresponding to each dataset (CUB: 200 classes, Flower: 102 classes, Aircraft: 100 classes).*
> > >
> > > As shown, ATHA consistently improves accuracy under both 100‑way and full‑label‑space settings, with only a modest increase in inference time. This clearly demonstrates that **the token selection mechanism is category‑agnostic** (it only depends on per‑patch maximum similarity to all class embeddings) and scales gracefully to large label spaces. The performance gain does not diminish as the number of categories increases.
> > >
> > > We hope these additional experiments and explanations address your concerns.

---

### Official Review · Reviewer_PX9i · 2026-03-13

**Soundness:** 3
**Presentation:** 2
**Significance:** 3
**Originality:** 3
**Overall Recommendation:** 4
**Confidence:** 3

**Summary:**

This work identifies a phenomenon that breaking the alignment of low-similarity tail tokens can improve model performance, and proposes the ATHA fine-tuning strategy based on this insight. The work challenges the traditional uniform vision-text alignment paradigm for CLIP adaptation, and through rigorous theoretical analysis and extensive experiments, verifies the effectiveness and superiority of the proposed method.

**Compliance With Llm Reviewing Policy:**

Affirmed.

**Key Questions For Authors:**

See Major Weakness.

**Limitations:**

yes

**Strengths And Weaknesses:**

Strengths:
1. The authors proposes an interesting idea that rather than forcing uniform alignment, pushing away tail tokens (image patch tokens with the lowest semantic similarity to target class texts) from textual embeddings, can consistently improve target domain performance.
2. Evaluate on four classic and challenging target domains with large domain shifts, which verifies the generalization ability of the method across different domains.
3. The method is simple, plug-and-play, and incurs negligible extra parameters, making it practical for real-world low-resource deployment.

Weaknesses:
1. A central concern is that indiscriminately discarding low-similarity tail tokens may remove not only noise but also potentially useful rare features, which could harm robustness and generalization. Additionally, the choice of cosine similarity​ as the sole metric for ranking token–text alignment is not justified with alternatives. It is unclear whether other similarity measures (e.g., Euclidean distance, learned metrics, mutual information) would yield more stable or better performance. A discussion of metric sensitivity and selection criteria​ is necessary.
2. The method is currently tested on CLIP-ViT/B-16. It remains unknown whether ATHA remains effective on larger CLIP encoders, which have stronger representational power but also a higher risk of overfitting in few-shot settings.
3. In the ChestX domain, the absolute improvement is modest and much smaller than in other domains. The paper acknowledges that in extreme domain shifts or tasks with very fine inter-class differences, the benefit of ATHA may diminish, but it does not provide a deeper analysis​ of why this happens or suggest targeted improvements​.
4. A portion of the reported gains may come from the strong zero-shot capability of the pre-trained CLIP model​ itself, rather than from the ATHA design. The paper does not include equal-parameter or equal-computation comparisons with full fine-tuning of CLIP, so it is impossible to tell whether the structural innovation is truly the main driver of performance, or whether the method simply makes good use of the frozen backbone’s capacity. Such a comparison is essential to establish whether ATHA would still be advantageous if more parameters are allowed.
5.The manuscript needs further polishing: there are several typos (e.g., “AHTA”) and at least one orphan sentence fragment​ (e.g., the standalone phrase at lines 192 and 256).

---

> ### Author Rebuttal · Authors · 2026-03-31
>
> Thank you for your thorough review and constructive feedback. Below are our detailed responses to your concerns:
>
> ### **W1. On discarding tail tokens & similarity metric**
>
> We appreciate this insightful concern. We clarify that ATHA does **not discard** tail tokens; it **pushes them away** from the text embeddings while preserving their representation.
>
> To validate that useful information concentrates in mid‑ and high‑similarity tokens, we vary the tail token ratio. Performance improves from 0 to 0.1, confirming that tail tokens are mostly noise, but declines beyond 0.2, indicating that informative tokens are also included in the tail set and suppressed. Thus, the optimal ratio of 0.1 effectively suppresses noise while preserving useful signals.
>
> |Tail tokens ratio|0 (baseline)|0.1|0.2|0.4|0.6|0.8|
> |-|-|-|-|-|-|-|
> |Ave. Performance|65.99|**68.53**|67.28|66.28|66.24|65.86|
>
> We compare cosine similarity with alternatives. All metrics achieve similar performance, showing that ATHA is not sensitive to the choice of similarity measure.
>
> |Distance metrics|CropDisease|EuroSAT|ISIC|ChestX|Ave.|
> |-|-|-|-|-|-|
> |Baseline|96.21|92.52|51.10|24.13|65.99|
> |Euclidean distance|97.55|93.42|56.27|26.31|68.39|
> |Learned metrics|97.41|93.22|56.59|26.61|68.46|
> |Mutual information|97.22|93.44|55.92|27.04|68.41|
> |**Cosine similarity**|**97.62**|**93.41**|**56.42**|**26.67**|**68.53**|
>
> ### **W2. On evaluation with larger CLIP encoders**
>
> We have conducted additional experiments on larger backbones under the 1-shot setting. The results confirm that ATHA consistently improves performance across larger encoders.
>
> |Backbone|CropDisease|EuroSAT|ISIC|ChestX|Ave.|
> |-|-|-|-|-|-|
> |CLIP (ViT-B/16) (~126M)|85.32|81.41|35.23|21.73|55.92|
> |**+Ours**|**87.99**|**82.56**|**38.86**|**24.00**|**58.35**|
> |CLIP (ViT-L/14) (~396M)|85.76|79.49|33.03|21.30|54.90|
> |**+Ours**|**86.82**|**80.07**|**38.03**|**22.67**|**56.90**|
> |SigLIP2 (~204M)|80.99|68.09|28.90|21.33|49.83|
> |**+Ours**|**84.80**|**71.98**|**32.67**|**22.97**|**53.11**|
> |PE-Core (~448M)|87.14|77.80|39.34|21.97|56.61|
> |**+Ours**|**89.81**|**80.33**|**40.52**|**22.48**|**58.29**.|
>
> ### **W3. On the modest improvement in the ChestX domain**
> We acknowledge the reviewer's observation that ChestX shows smaller absolute gains compared to other domains.  However, as shown in the table below, **all existing CLIP-based methods struggle on ChestX** due to its inherent difficulty：
>
> |Model|ChestX|
> |-|-|
> |Baseline|24.13|
> |+CC-CDFSL (CVPR26)|25.47|
> |+VtT (CVPR26)|26.42|
> |+SVL (CVPR26)|26.61|
> |**+Ours**|**26.67**|
>
> 1. **ChestX is inherently challenging with low baseline performance.** Recent benchmarks confirm that VLMs struggle with medical images compared to natural images (Zhang et al., MICCAI 2024).
> 2. **Cross-modal alignment is severely disrupted.** Prior works (Zhang et al., CVPR 2026) claim that CLIP’s vision-text alignment is fundamentally broken under extreme domain shifts in ChestX, where cosine similarities between patches and class texts are often negative, making meaningful alignment difficult.
> 3. **Alternative architectures better capture ChestX’s fine-grained local patterns.** CNN-based models and DINO-ViT inherently outperform CLIP on medical imaging due to their local inductive biases and texture-focused representations, while CLIP’s global semantic alignment objective is ill-suited for localized pathologies  (Chen et al., CVPR 2024).
>
> Given these challenges, ATHA still achieves the best performance among CLIP-based SOTAs.
>
> ### **W4. On the contribution from CLIP's zero-shot capability vs. ATHA**
>
> Recent studies (Zhou et al., IJCV 2022)  have shown that fully fine‑tuning CLIP in few‑shot scenarios often leads to overfitting, while PEFT methods generally achieve better generalization. We compare full fine‑tuning with other methods, and the results confirm that **keeping most CLIP parameters frozen is more effective**. Among all compared SOTAs, **ATHA achieves the best performance**.
>
> |Model|Ave. Performance|Learnable Params|
> |-|-|-|
> |Baseline-LoRA|65.99|~1.5M|
> |Baseline-Full finetuning|62.63|~124M|
> |+CC-CDFSL (CVPR26)|67.90|~2M|
> |+SVL (CVPR26)|68.38|~1.5M|
> |**+Ours**|**68.53**|**~1.5M**|
>
> To further validate that ATHA’s improvement is **not tied to a specific fine‑tuning method or parameter budget**, we integrate it with three representative PEFT strategies. ATHA consistently improves performance across all, regardless of their learnable parameter counts.
>
> |Strategy|Learnable Params|CropDisease|EuroSAT|ISIC|ChestX|Ave.|
> |-|-|-|-|-|-|-|
> |CoOp|~2K|91.88|83.22|43.36|22.69|60.29|
> |**+Ours**|~2K|**93.33**|**83.81**|**46.67**|**24.32**|**62.03**|
> |MaPLe|~0.9M|96.22|90.80|50.97|24.12|65.53|
> |**+Ours**|~0.9M|**96.93**|**92.02**|**55.91**|**25.49**|**67.59**|
> |LoRA|~1.5M|96.21|92.52|51.10|24.13|65.99|
> |**+Ours**|~1.5M|**97.62**|**93.41**|**56.42**|**26.67**|**68.53**|
>
> ### **W5. On presentation and typos**
>
>  We promise we will fix typos and grammatical errors in the final version.

---

> > ### Author Rebuttal · Reviewer_PX9i · 2026-04-08
> >
> > My concerns have been addressed and I will keep my score.

---

### Official Review · Reviewer_NYBP · 2026-03-13

**Soundness:** 3
**Presentation:** 2
**Significance:** 3
**Originality:** 3
**Overall Recommendation:** 5
**Confidence:** 3

**Summary:**

This paper proposes Adaptive Tail-Head Alignment (ATHA), a novel fine-tuning strategy for CLIP in the setting of source-free cross-domain few-shot learning (SF-CDFSL). The key insight is that not all image patch tokens should be aligned to class text embeddings equally during adaptation. The method ranks patch tokens based on their similarity to class text features and applies differential alignment: tokens with high similarity ("head tokens") are further aligned toward the corresponding class embedding, while tokens with very low similarity ("tail tokens") are pushed away.

The method is motivated by the idea that many low-similarity tokens represent noise or domain-specific background information, which may harm adaptation when forced to align with class semantics. By strengthening alignment for informative tokens while suppressing alignment for uninformative ones, ATHA aims to improve generalization under domain shift with limited target-domain supervision.

The paper provides a detailed description of the proposed method and evaluates it on several cross-domain few-shot benchmarks. The results show that ATHA consistently improves performance over strong baselines such as CLIP-LoRA and other recent CDFSL methods. The authors also present extensive ablation experiments and analysis, including comparisons with loss-based alignment formulations, demonstrating that direct feature modulation yields stronger improvements.

**Compliance With Llm Reviewing Policy:**

Affirmed.

**Final Justification:**

The additional experiments cleanly answered lingering questions about computational cost and applicability to other backbones. With these experimental results, I believe that the paper makes a compelling argument for use of the method.

**Key Questions For Authors:**

1. What is the cost for training and inference relative to other methods?
2. I would be curious if it is possible to see results on other backbones but I understand if that is not possible during the rebuttal period and  so lack of results will not affect my final score.

**Limitations:**

yes

**Strengths And Weaknesses:**

# Strengths
## Soundness
The paper is technically sound and is supported by empirical evidence. The experiments are well-designed and ablations demonstrate the value of method components and variants. In particular, the analysis of why a loss-based formulation preempts questions that may arise during reading.

## Presentation
Overall, information is presented well and the paper is easy to read.

## Significance
This paper addresses an important problem and achieves strong results relative to baselines.

## Originality
The idea of using adaptive alignment for high- and low-similarity tokens in this context is novel and provides interesting insights into CLIP behavior. The paper sufficiently compares to similar existing approaches.

# Weakness
## Soundness
N/A

## Presentation
1. There are small typos and grammar errors throughout. Copy editing is recommended.

## Significance
1. What is the computational overhead of this method? This should be included for both training and inference and relative to other methods.

## Originality
N/A

---

> ### Author Rebuttal · Authors · 2026-03-31
>
> Thank you for your thorough review and constructive feedback. Below are our detailed responses to your questions:
>
> ### **W1. Presentation Improvements**
>
> We sincerely appreciate your note on the presentation. We promise we will carefully fix typos and grammatical errors in the final version. We appreciate your patience.
>
> ### **W2&Q1. Computation cost**
>
> We compare our method with recent SOTAs in the table below. Our method introduces **only 24 additional parameters** (<0.00002% increase), and achieves the best average performance while introducing **the smallest training overhead among all SOTA methods**. Inference time remains nearly unchanged, demonstrating an excellent trade‑off that makes ATHA practical for real‑world deployment.
>
> |Model|Mark|Train (s/epoch) ↓|Infer (ms/sample) ↓| Learnable Params ↓|Average Performance ↑|
> |-|-|-|-|-|-|
> |Baseline|-|29.63|215.36|~1.50M|65.99|
> |+CC-CDFSL|CVPR26|86.84|215.36|~2.03M|67.90|
> |+SVL|CVPR26|50.56|215.36|~1.50M|68.38|
> |**+Ours**|Ours|**32.64**|219.60|**~1.50M**|**68.53**|
>
> ### **Q2. Experiments with other backbones**
>
> While our primary experiments use CLIP (ViT-B/16), we conduct additional experiments on CLIP (ViT-L/14), SigLIP2, and PE-Core under the 1-shot setting. These results demonstrate that ATHA is **backbone‑agnostic** and generalizes well across different vision‑language models.
>
> |Backbone|CropDisease|EuroSAT|ISIC2018|ChestX|Ave.|
> |-|-|-|-|-|-|
> |CLIP (ViT-L/14)|85.76|79.49|33.03|21.30|54.90|
> |**+Ours**|**86.82 (+1.06)**|**80.07 (+0.58)**|**38.03 (+5.00)**|**22.67 (+1.37)**|**56.90 (+2.00)**|
> |SigLIP2|80.99|68.09|28.90|21.33|49.83|
> |**+Ours**|**84.80 (+3.81)**|**71.98 (+3.89)**|**32.67 (+3.77)**|**22.97 (+1.64)**|**53.11(+3.28)**|
> |PE-Core|87.14|77.80|39.34|21.97|56.61|
> |**+Ours**|**89.81 (+2.67)**|**80.33 (+2.53)**|**40.52 (+1.18)**|**22.48 (+0.51)**|**58.29 (1.68)**|
>
> ### **Further validation of effectiveness**
>
> To further demonstrate ATHA’s strong generalization, beyond the additional backbones evaluated in the previous section, we further provide results in combination with existing fine‑tuning strategies and on different datasets.
>
> **(1) Combination with finetuning strategies**
>
> We also validate that ATHA can be orthogonally combined with existing prompt tuning methods (CoOp, MaPLe, Lora) to further boost performance, demonstrating its flexibility.
>
> |Strategy|CropDisease|EuroSAT|ISIC2018|ChestX|Ave.|
> |-|-|-|-|-|-|
> |CoOp|91.88|83.22|43.36|22.69|60.29|
> |**+Ours**|**93.33 (+1.45)**|**83.81 (+0.59)**|**46.67 (+3.31)**|**24.32 (+1.63)**|**62.03 (+1.74)**|
> |MaPLe|96.22|90.80|50.97|24.12|65.53|
> |**+Ours**|**96.93 (+0.71)**|**92.02 (+1.22)**|**55.91 (+4.94)**|**25.49 (+1.37)**|**67.59 (+2.06)**|
> |LoRA|96.21|92.52|51.10|24.13|65.99|
> |**+Ours**|**97.62 (+1.41)**|**93.41 (+0.89)**|**56.42 (+5.32)**|**26.67 (+2.54)**|**68.53 (+2.54)**|
>
> **(2) More dataset results**
>
> To further validate robustness, we extend evaluation to additional datasets under both 1-shot and 5-shot settings, including **Stanford Cars** (fine‑grained vehicle classification), **DTD** (texture recognition), **RUOD** (underwater object with severe visual degradation), and **WEDGE** (weather‑degraded imagery). Consistent improvements are observed across all datasets, demonstrating ATHA’s strong generalization to diverse domains.
>
> |Method|Shot|WEDGE|RUOD|DTD|Stanford Cars|Ave.|
> |-|-|-|-|-|-|-|
> |Baseline|1|71.13|77.41|81.74|92.77|80.76|
> |**+Ours**|1|**74.68 (+3.55)**|**79.77 (+2.36)**|**82.80 (+1.06)**|**93.56 (+0.79)**|**82.70 (+1.94)**|
> |Baseline|5|80.93|88.43|89.29|97.35|89.00|
> |**+Ours**|5|**84.17 (+3.24)**|**89.52 (+1.09)**|**89.96 (+0.67)**|**98.09 (+0.74)**|**90.44 (+1.44)**|
>
>
>
> We appreciate your insightful feedback and hope these revisions address your concerns. Thank you again for your time and effort in reviewing our work!

---

> > ### Author Rebuttal · Reviewer_NYBP · 2026-04-03
> >
> > I am satisfied with the additional experiments and I will raise my score to a 5. Thank you to the authors for addressing my concerns.

---

> > > ### Author Response · Authors · 2026-04-04
> > >
> > > Thank you again for your recognition of our work. We will carefully incorporate your suggestions in the revised version. We truly appreciate your time and expertise.

---

### Decision · Program_Chairs · 2026-04-30

**Decision:**

Accept (regular)

**Comment:**

Current fine-tuning methods for Vision-Language Models typically align all image tokens with text descriptions to adapt to new domains. However, under large domain shifts with scarce data, forcing irrelevant "tail tokens" to align with texts leads to severe overfitting. This paper introduces Adaptive Tail-Head Alignment (ATHA), a novel framework that dynamically strengthens the alignment of semantically relevant tokens while actively pushing away irrelevant ones. This asymmetric token modulation effectively prevents overfitting and improves generalization in cross-domain scenarios without requiring source data.

The reviewers found the idea interesting and well motivated, the procedures simple but effective, the analysis solid, and the results convincing. They raised several questions that the rebuttal addressed convincingly.  Overall, the papers was considered a solid contribution to ICML